# A Synopsis of Hepatitis C Virus Treatments and Future Perspectives

Christian Medina [1], Alexis Hipólito García [1,*] , Francis Isamarg Crespo [1], Félix Isidro Toro [1], Soriuska José Mayora [1] and Juan Bautista De Sanctis [2,3,*]

[1] Institute of Immunology Dr. Nicolás E. Bianco C., Faculty of Medicine, Universidad Central de Venezuela, Caracas 1040, Venezuela; cmedima1108@gmail.com (C.M.); drafranciscrespo@gmail.com (F.I.C.); torfelix@gmail.com (F.I.T.); soriuskamayora@gmail.com (S.J.M.)

[2] Institute of Molecular and Translational Medicine, Faculty of Medicine and Dentistry, 779 00 Olomouc, Czech Republic

[3] The Czech Advanced Technology and Research Institute (Catrin), Palacky University, 779 00 Olomouc, Czech Republic

\* Correspondence: alexisgarcia27@gmail.com (A.H.G.); juanbautista.desanctis@upol.cz (J.B.D.S.)

**Abstract:** Hepatitis C virus (HCV) infection is a worldwide public health problem. Chronic infection with HCV can lead to liver cirrhosis or cancer. Although some immune-competent individuals can clear the virus, others develop chronic HCV disease due to viral mutations or an impaired immune response. IFNs type I and III and the signal transduction induced by them are essential for a proper antiviral effect. Research on the viral cycle and immune escape mechanisms has formed the basis of therapeutic strategies to achieve a sustained virological response (SVR). The first therapies were based on IFNα; then, IFNα plus ribavirin (IFN–RBV); and then, pegylated-IFNα-RBV (PEGIFNα-RIV) to improve cytokine pharmacokinetics. However, the maximum SVR was 60%, and several significant side effects were observed, decreasing patients' treatment adherence. The development of direct-acting antivirals (DAAs) significantly enhanced the SVR (>90%), and the compounds were able to inhibit HCV replication without significant side effects, even in paediatric populations. The management of coinfected HBV–HCV and HCV–HIV patients has also improved based on DAA and PEG-IFNα-RBV (HBV–HCV). CD4 cells are crucial for an effective antiviral response. The IFNλ3, IL28B, TNF-α, IL-10, TLR-3, and TLR-9 gene polymorphisms are involved in viral clearance, therapeutic responses, and hepatic pathologies. Future research should focus on searching for strategies to circumvent resistance-associated substitution (RAS) to DAAs, develop new therapeutic schemes for different medical conditions, including organ transplant, and develop vaccines for long-lasting cellular and humoral responses with cross-protection against different HCV genotypes. The goal is to minimise the probability of HCV infection, HCV chronicity and hepatic carcinoma.

**Keywords:** hepatitis C virus; chronic HCV; antivirals; sustained virological response; IFN therapy; vaccines

## 1. Introduction

Hepatitis C virus (HCV) was discovered by Harvey J. Alter, Michael Houghton and Charles M. Rice (Nobel Prize in Medicine and Physiology in 2020). Their research was crucial to identifying and characterising the viral genome, developing serological and molecular diagnostic methodologies, and providing fundamental bases for studying the virus's pathophysiology [1,2].

### 1.1. Viral Genome

Structurally, HCV is a positive-stranded RNA virus with a lipoprotein envelope that presents a spherical structure of approximately 55 nm in diameter and is taxonomically located within the genus Hepacivirus of the family Flaviviridae [1–3]. The genome is approximately 9.6 kb long and codes for a polyprotein of about 3010 amino acids that is

proteolytically processed by viral and cellular enzymes to generate at least 10 proteins (Figure 1). These proteins include three "structural" polypeptides: (1) the nucleocapsid or "core" protein (C) and two envelope proteins (E1 and E2); (2) two proteins that are essential for virion production (p7 and NS2); and five non-structural proteins that are essential for viral replication (NS3, NS4A, NS4B, NS5A and NS5B) (Figure 1) [1–3].

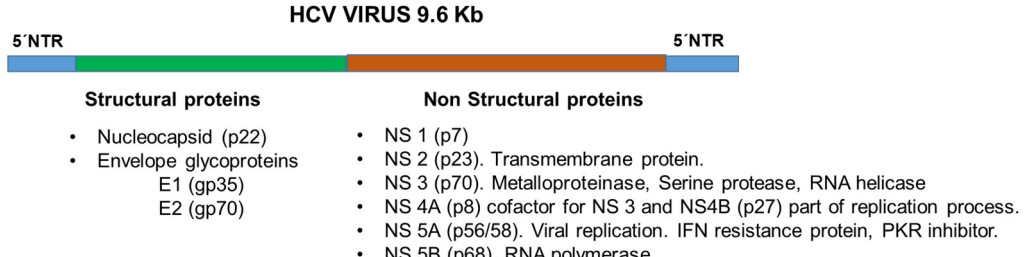

**Figure 1.** HCV genomic virus structure and viral proteins.

### 1.2. Viral Transmission and Viral Receptors

HCV RNA has been successfully detected in blood (including serum and plasma), saliva, tears, seminal fluid, ascitic fluid and cerebrospinal fluid [2–6]. Around 185 million people are infected with HCV [4,5]. In particular, HCV infections have been linked to intravenous drug abuse or poor medical practices in haemodialysis, transfusion of haemo components and blood products in resource-limited areas of the world. The frequency of perinatal and sexual transmission is low [2–6]. According to estimates by the Center for Disease Control (CDC) for 2013, HCV infection accounted for the highest number of deaths and the mortality rate (5.0 deaths/100,000 population) among hepatitis of viral origin [7,8]. The number of deaths from HCV infection in 2012 compared to 2015 increased from 18,650 to 19,629 in the US [7,8].

Viral transmission requires that infectious virions come into contact with susceptible host cells through specific and co-receptors or non-specific receptors. The key receptors for HCV entry are CD81 [9], the scavenger receptor class B member 1 (SCARB1) [10], the proteins of the tight junction, claudin-1 (CLDN1) and occludin (OCLN) [11], EGFR (epidermal growth factor receptor) and EPH receptor A2 (ephrin type-A receptor 2), involved in HCV virus entry and, probably, in oncogenic transformation [12]. Other receptors that are not highly specific but are involved in virus entry and escape are the very low-density lipoprotein receptor (VLDL-R), the LDL receptor (LDL-R) [13,14], Niemann-Pick C1-like 1 receptor (NPC1L1) [15], heparan sulphate proteoglycan (HSPG) [16] and the Fc receptors by immune complex [17]. Due to the variety of receptors and co-receptors, non-hepatic cells, including leukocytes, are infected or their function is affected by the viral infection [18–20]. The decrease in neuroinflammation, decreased brain volume and increased evoked potentials in HCV-infected individuals treated with IFN-free therapy suggest that other extrahepatic manifestations should be studied thoroughly [21,22].

### 1.3. Antiviral Response and Immune Response Activation

HCV has an enormous replicative capacity, reaching 105–107 IU/mL titres in the first days of infection [3,6]. Only a minority of those infected particles are spontaneously eliminated by the immune response [23,24]. The efficiency depends on the virus's heterogeneity, mechanisms of immune response evasion, and host-specific factors such as age, race, sex and genetic markers [5,24–26]. During the initial response, the host's immune system reacts similarly to other viral infections by inducing the transcription and secretion of type I and III interferons to restrict viral replication [5,6,23]. The IFN signalling cascade includes Janus kinases (JAK), the STAT (signal transducers and activators of transcriptions family) transcription factors that dimerise and translocate from cytosol, and the IRF (IFN-stimulated gene factor), which binds to the IFN-stimulated response elements 3 and 9 (ISREs) gene promoters, leading to the transcription of numerous IFN-stimulated genes (ISGs) [27]. The HCV viral core

protein interferes with STAT signalling by decreasing STAT1 accumulation and promoting its proteosome-dependent degradation, affecting the IFN-λ signalling pathway and favouring HCV replication [28,29]. IFNα activates STAT2, which also affects HCV replication [30]. However, IL-8, which is induced upon the viral infection, can facilitate viral escape by suppressing IFNα signalling (antagonism) [31].

Other critical players in the cellular immune response against HCV are (1) the blockade of protein kinase R (PKR) phosphorylation and dimerisation by HCV proteins [32,33]; (2) the expression of human myxovirus resistance protein A (MxA), which inhibits HCV replication by activating the JAK-STAT pathway independently of IFNα [34]; and (3) IFN signalling is enhanced in females, especially at a young age, compared to males, with the effect seemingly being dependent on oestradiol levels [35,36]. In the early stages, HCV clearance occurs predominantly via non-cytolytic effector mechanisms induced by the IFN-stimulated genes (ISG) in hepatocytes [5,6,24,25]. Nevertheless, these signals may be insufficient to decrease the viral burden during the initial stage.

After 4 to 8 weeks, HCV-specific T lymphocytes are recruited to the liver in the second or late phase of acute hepatitis. This phase lasts 4 to 10 weeks and is the best opportunity for the immune system to clear the virus [5,6,24,25]. However, MHC antigen presentation can be suppressed by NS4A/B virus proteins in infected cells [37,38], and the HCV core proteins NS3, NS5A, and NS5B can activate apoptosis of mature dendritic cells [37,38], inhibiting the innate response and decreasing antigen presentation [24,37,38]. The absence of antigens results in the lack of recognition of infected cells [24,37,38]. The peripheral NK cell cytotoxic response in HCV patients is usually impaired [20], suggesting that tissue NK cells could not eliminate the virus; peripheral T-cell responses were unaffected [37–39]. As shown in coinfected HIV–HCV patients, the CD3CD4 cell population may be crucial in virus clearance [39,40]. Viral clearance is achieved in one out of three infected patients and requires a sustained and prolonged specific $CD^{4+}$ and $CD^{8+}$ T lymphocyte response against different HCV proteins [23,24,39–41]. Moreover, the gene polymorphisms of the host, mainly IFNλ3, IFNλ4, IL28B, inflammatory cytokines (IL-12, TNFα), IL-10, and Toll-like receptors, especially in their role in coinfection with HBV and HIV, are crucial in the efficiency of viral clearance [38]. Viral escape and chronicity can be detected via the lack of expression of proteins induced by ISGs; the virus effectively blocks the IFN signal pathway [4,24–26,42]. Subjects with chronic hepatitis have a 25% higher risk of developing liver cirrhosis and hepatocarcinoma after 10–40 years of infection due to the continuous necrosis and inflammation of the liver [4,6]. It is the second most common cause of liver cancer worldwide [6,26].

The virus's gene variability is mainly due to the low corrective activity in the RNA polymerase (NS5B) responsible for viral genome replication, generating multiple variants [6,25,43–45]. According to the current classification, seven HCV genotypes are identified based on the nucleotide variability in the genome sequences analysed from various geographic regions [6,25,43–47]. Genotypes 1 to 4 vary in distribution and prevalence depending on the geographic area [6,25,43–47]. Genotype 1 is the most common in the United States, Latin America and Europe, accounting for 46% of all infections, followed by genotypes 3 (22%) and 2 and 4 (13% each). Around 40% of all infections in Asia are genotype 3, and genotype 4 is the most common (71%) in North Africa. Patients infected with genotype 1 have a lower therapeutic response than those infected with the other genotypes [6,43–47].

### 1.4. Coinfections: HBV–HCV and HCV–HIV

HCV coinfection with HBV and/or HIV has also been described to affect the immune response, viral burden and chronicity [6,48–50]. HBV–HCV coinfection is common in some endemic regions, and it is often difficult to establish since patients have undetectable levels of antibodies against HBV surface proteins but have detectable levels of HBV DNA (a consistent marker of active replication) [50]. HCV infection can activate IFN type I and III synthesis, decreasing HBV replication; however, if HCV downregulates IFN I and III signalling, it facilitates liver infection by both viruses [49,50]. The severity of

the coinfection also depends on the genotype of the HCV virus and the host's immune response [49,50]. Viral replication is very high in immunosuppressed or immune-deficient patients and requires special attention [50]. Coinfected individuals have a higher incidence of liver disease and hepatocellular carcinoma [48–50].

Given the modes of transmission of the blood-borne viruses HIV, HBV and HCV, coinfection with two or three viruses is highly probable in endemic areas [48–50]. The risk of multimorbidity is higher in injection drug users; these patients have a three-fold higher risk of developing hepatic disease than those infected only with HCV [48–50]. The viral load of all the viruses involved may be enhanced in coinfection, impairing viral clearance due to an inefficient immune response [48–50]. In the three viral infections, effective CD4 cells, in addition to IFNs type I and III, are crucial for virus clearance. Lower numbers of circulating CD4 facilitate the viral escape of these viruses [48–50]. Even though CD8 cells are critical to eliminating viral-infected cells, recent reports revealed that CD8 is involved in liver damage [48–52]. The increase in primed CD8 cells is probably responsible for the augmented liver damage in coinfected individuals [48–52].

The treatment of HCV infection has represented a real challenge in therapeutics to develop treatments capable of generating a sustained virological response (SVR) (Figure 1). An SVR is achieved when HCV RNA is no longer detectable in the blood after 12 weeks of therapy, with decreased antibody titres and improved liver pathology. HCV reinfection rarely occurs; nonetheless, the infection of non-hepatic cells may facilitate viral escape [4–6]. The development of preventive HCV vaccines remains another primary strategy to eliminate the disease globally. The extreme genetic diversity of HCV represents a well-known obstacle to developing an effective vaccine [43].

## 2. Interferon α and Ribavirin

Interferons do not have a unique mechanism of action. The intracellular signals, second messengers and proteins induced by IFNs are responsible for the antiviral activity [38,42,53–55]. Virus resistance, as described before, refers to the inhibition of these signalling events, directly or indirectly, by the virus [53–55].

IFNα therapy was first tested by Hoofnagle et al. [56] in a study showing decreased levels of aminotransferases in 8 out of 10 chronically HCV-infected patients. Another investigation of 44 German patients infected with HCV and treated with IFNα2b showed undetectable serum HCV RNA levels and normal alanine aminotransferase levels in 98% of patients after 24 weeks of treatment [56].

To improve the standard IFNα2b therapy, the drug ribavirin (RBV), a guanosine nucleoside analogue with antiviral activity, was introduced in the late 1990s [57,58]. This drug was adjusted to the body mass for 24 or 48 weeks, ranging from 1000 to 1200 mg orally daily. The SVR was much higher with the combined IFN–RBV therapy compared to the standard treatment consisting of IFN alone [59,60].

The mechanisms of action of RBV have not yet been fully elucidated. Nevertheless, four possible mechanisms have been proposed: (1) antiviral effect against HCV RNA polymerase-dependent RNA, (2) depletion of the intracellular pool of guanosine triphosphate (GTP), (3) induction of the misincorporation of nucleotides by viral RNA polymerase, and (4) alteration of the cytokine balance from a Th2-type to a Th1-type profile with antiviral properties [61].

In the early 2000s, the structure of IFN was modified by adding polyethene glycol (PEG) chains [62–68]. Pegylation of IFN (PEGIFN) confers constant absorption, a longer half-life in serum and lower systemic clearance. These changes allow sustained serum concentrations and improve the SVR when combined with RBV [62–68]. PEGIFN has two forms of presentation in HCV treatment: pegylated IFNα2a (PEG IFNα2a) and pegylated IFNα2b (PEG IFNα2b). PEG IFNα2a produces a higher SVR than IFN PEGα2b when combined with RBV, achieving a cure rate between 56 and 54% and fewer secondary effects [67,68]. A randomised study by Manns et al. reported the advantage of PEGIFN treatment with RBV compared to IFNα2b monotherapy [68].

The standard treatment for many years for HCV infection was PEGIFN/RBV; nonetheless, the side effects led to a decrease in adherence to the treatment in the early stages [69]. The most relevant side effects described were (1) anaemia, with around 54% of treated patients reported a reduction in haemoglobin of $\geq 3$ g/dL; (2) neutropenia and thrombocytopenia, leading to compensatory drugs such as granulocyte colony-stimulating factor (G-CSF) and thrombopoietin receptor agonists (eltrombopag) being used in these patients [69–72]; and (3) chronic fatigue syndrome and psychiatric symptoms such as depression [69–72]. Given the poor adherence to treatment with PEGIFN/RBV and the low percentage of the SVR, particularly against the HCV genotype 1, other therapeutic tools have been introduced after IFN (Figure 2).

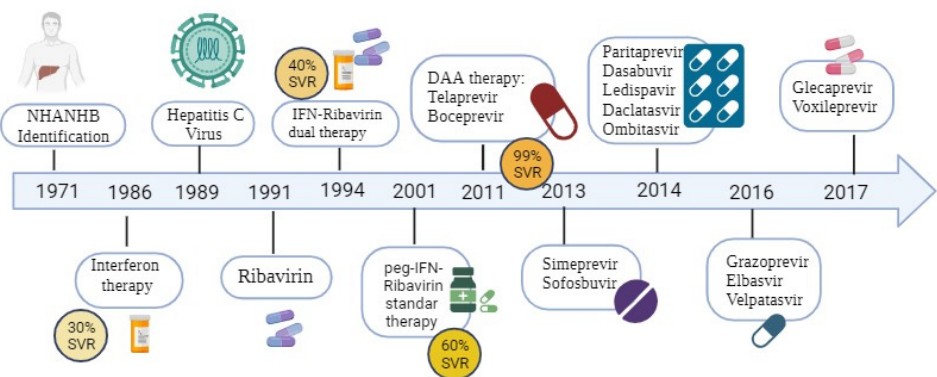

**Figure 2.** History of advances in treatment of chronic HCV infection and evolution of sustained viral response. NANBH: non-A, non-B hepatitis; IFNα: interferon α; Peg-IFN: pegylated interferon-α. Over the years, there has been a marked improvement in the HCV clearance efficiency.

## 3. Direct-Acting Antivirals (DAA)

Viral proteases are considered important targets for antiviral therapy [73–75]. NS3/4 proteases, the multifunctional protein NS5A, and the viral polymerase NS5B have been the most studied potential therapeutic targets in HCV [73–75]. A range of drugs, direct-acting antivirals (DAAs), have been developed against HCV proteases. The future aim is to find more specific and compelling compounds with few side effects [73–75].

### 3.1. NS3/4 Protease Inhibitors

Proteolytic cleavages at the hepatitis C virus (HCV) polyprotein generating NS3-NS4A, NS4A-NS4B, NS4B-NS5A, and NS5A-NS5B are produced by the virus-encoded serine protease in NS3 [76,77]. The enzyme is specific; it uses an extended polydentate-binding cleft with several recognition subsites [76,77]. This polypeptide forms a heterodimeric complex with the NS4A protein, an essential cofactor that induces proteolytic activity and helps anchor the heterodimer to the endoplasmic reticulum [74–77].

The first DAAs were designed based on enzyme active sites [77]. However, HCV proteases require a large peptide substrate, with which the enzyme establishes multiple weak interactions distributed over an extended surface area. The requirement for large substrates was a significant concern when developing orally bioavailable small-molecule drugs [74,77].

Ciluprevir was the first NS3 protease inhibitor developed against HCV [74,77]. It demonstrated an early viral load reduction in humans and established the proof of concept for future HCV protease inhibitors. The development of ciluprevir was discontinued due to cardiac toxicity observed in monkeys at high doses [74,77].

### 3.1.1. Telaprevir and Boceprevir

In 2011, telaprevir was the first DAA approved by the US Food and Drug Administration (FDA). As an NS3/4A protease inhibitor, it was designed to treat viral infection by genotype 1 or patients previously treated with IFN–RBV [78]. The telaprevir/IFN/RBV

combination offered an SVR between 60 and 75% [78]. The recommended dose of this drug was 750 mg three times a day for 12 weeks. Nonetheless, this schedule generated considerable side effects, including pruritus, rash, anaemia, nausea, vomiting and anal discomfort. Skin erythema led to the discontinuation of treatment in 4% of patients. Severe skin reactions of the Stevens–Johnson syndrome type have also been reported and were the most critical limitation of the use of telaprevir [78].

In 2011, boceprevir, another NS3/4A protease inhibitor approved by the European Medicines Agency for HCV treatment, was introduced [79]. This drug was co-administered with PEGIFN-RBV and was only effective in patients infected with viral genotype 1, with an SVR of 67–68% [79]. The dose of boceprevir was 800 mg every 8 h, and the duration of treatment depended on the response, ranging from 24 to 48 weeks [80]. Therapy was discontinued in all patients with HCV RNA levels above 100 IU at week 12 or if viral RNA was present at week 24, as an SVR is rarely achieved under these circumstances and prevents the development of resistance to boceprevir. The main adverse effect attributed to this DAA is anaemia, so these patients were prescribed erythropoietin during treatment [75].

Both boceprevir and telaprevir have been classified as category B drugs in pregnancy, i.e., with no evidence of risk in humans. However, both drugs are currently discontinued due to the emergence of more effective and pangenotypic DAAs [75].

### 3.1.2. Asunaprevir

Asunaprevir is an efficient N3/4A protease inhibitor with favourable liver distribution used as a co-treatment with IFNPRG-RBV or other DAAs in patients infected with HCV genotypes 1 to 4 [81]. However, the drug's pharmacodynamics are markedly affected by using drugs that inhibit the organic anion-transporting polypeptide, like rifampin, mild hepatic impairment and Asiatic ethnic group [75]. A new scheme, asunaprevir, is used in China, Japan and parts of Europe using daclastavir and beclabuvir (an indolobenzapine-like allosteric inhibitor of HCV polymerase). This combination, administered for 12 weeks, achieves an SVR ≥ 90%, regardless of the cirrhosis stage, previous use of RBV or IFN failure [82]. Despite its high SVR rate, asunaprevir is limited to a twice-daily dosing schedule, is associated with safety concerns involving patients with advanced liver and kidney disease, and exhibits minimal potency against HCV genotypes outside of genotype 1. Its use should be an additional option for specific patient subgroups. However, resistance-associated mutations were observed in the coinfected HCV/HIV patients, and changing its use for another DAA has been suggested [83].

### 3.1.3. Simeprevir

In 2013, simeprevir, the first single daily dose DAA (50 to 200 mg), was approved for treating HCV genotype 1 in the US and genotypes 1 and 4 in the European Union [84]. Simeprevir was approved for use with PEGIFN + RBV, although it was subsequently approved as combination therapy with sofosbuvir, with SVR rates of 79–100% reported after 12–24 weeks of treatment. Combining simeprevir and sofosbuvir represented the first completely IFN-free oral combination therapy for treating HCV genotype 1 [85]. Adverse reactions with simeprevir are less common than first-generation DAAs. The most frequent was rash (photosensitivity), followed by hyperbilirubinaemia, pruritus and nausea. Like other drugs, simeprevir is a DAA that has been discontinued because it has shown efficacy in only two HCV genotypes [75].

### 3.1.4. Paritaprevir

In 2014, paritaprevir was approved by the FDA in a fixed-dose co-formulated combination (FDC) with ritonavir (an HIV protease inhibitor) and ombitasvir (an NS5A inhibitor) for treating HCV genotype 4 [86]. For HCV genotype 1, the combination of paritaprevir/ritonavir/ombitasvir with dasabuvir (a non-nucleoside polymerase inhibitor) is known as PrOD [87]. This combination does not require dose adjustment for patients with renal impairment but is contraindicated in patients with severe hepatic dysfunction due to the

risk of potential toxicity. Side effects of PrOD with or without RBV are rare but include fatigue, nausea, pruritus, other skin reactions, insomnia and asthenia. In phase 3 trials of PrOD with or without RBV, less than 1% of subjects discontinued treatment due to a side effect. In the latest treatment guidelines of the American Association for the Study of Liver Diseases (AASLD) and the Infectious Diseases Society of America (IDSA), PrOD is no longer included in the treatment guidelines [75].

### 3.1.5. Grazoprevir

Over the years, better DAAs have been developed that exhibit fewer side effects. In 2016, the HCV NS3/4A protease inhibitor grazoprevir was approved by the FDA because studies showed that the kidneys excrete less than 1% of the drug, meaning that no dose adjustments were needed in patients with stage 4 and 5 chronic kidney disease [88]. Grazoprevir was combined with elbasvir (an NS5A inhibitor) for a combination therapy administered once daily for 12 weeks in patients infected with HCV genotypes 1b and 4 [88]. An SVR > 95% has been reported in patients treated with the combination of grazoprevir and elbasvir [89].

### 3.1.6. Glecaprevir and Voxileprevir

Glecaprevir, an NS3/4A protease inhibitor, is used in combination with pibrentasvir for individuals infected with all the genotypes (1 to 6) of HCV, without cirrhosis or compensated cirrhosis, and for those patients infected with HCV genotype 1 who have previously received an NS3/4A or NS5A DAA [90]. The reported adverse effects are mild, and the SVR is around 90%, depending on the genotype and previous treatment.

For treatment failure with first-generation DAAs, voxileprevir, an HCV NS3/4a protease inhibitor, is used alongside two previously approved drugs, sofosbuvir and velpatasvir, in a fixed-dose combination [91]. It demonstrated an SVR of 96–97% at 12 weeks of treatment in patients infected with genotypes 1–6 [91].

### *3.2. NS5B Polymerase Inhibitors*

### 3.2.1. Nucleotide: Sofosbuvir

Another class of DAAs developed for treating hepatitis C are the nucleotide analogues that inhibit the viral polymerase. These drugs bind with high affinity to the active site of the RNA-dependent RNA polymerase, the product of the NS5B gene, causing premature termination of the RNA strand and inhibiting HCV replication. Sofosbuvir was initially used in combination with ribavirin for the management of patients infected with genotype 2 (12 weeks), genotype 3 (24 weeks), and genotype 1 (24 weeks), and in combination therapy with IFN–PEG + RBV (12 weeks) for genotype 1. These regimens were associated with higher SVR rates than PEGIFN + RBV-only treatments, with no significant side effects reported by patients [92]. In 2014, the FDA approved using sofosbuvir and ledipasvir (an NS5BA inhibitor) for treating HCV genotype 1, 4, 5 or 6 infections [93]. Finally, in 2017, the combined use of sofosbuvir, voxilelprevir and velpatasvir was approved in those patients who showed treatment failure with other DAAs [91]. Additionally, sofosbuvir has been used with RBV to treat patients coinfected with HIV and HCV genotypes 1, 2 and 3, reporting high SVR rates [94].

### 3.2.2. Non-Nucleotide: Dasabuvir

Non-nucleotide analogue polymerase inhibitors bind outside the active site of NS5B RNA polymerase and cause allosteric inhibition of the enzyme. They generally have a lower barrier to resistance than nucleotide-analogue polymerase inhibitors. Dasabuvir is the first non-nucleotide analogue-type DAA approved in 2014 for treating HCV genotype 1 [95]. It is usually administered alongside paritaprevir/ritonavir/ombitasvir; it was effective in a group of patients [96]. However, dasabuvir is limited as a monotherapy due to its low genotypic coverage and the possibility of adding to other therapies [97].

### 3.3. NS5A Polymerase Inhibitors

NS5A inhibitors target the NS5A protein, which is essential for HCV RNA replication and viral assembly. The DAAs discussed below exhibit antiviral activity, primarily against genotype 1, with the activity varying against other genotypes. Combining a pangenotypic NS5A inhibitor with pangenotypic nucleotide analogue polymerase inhibitors currently represents the most widely used treatment against HCV [98].

### 3.3.1. Ledipasvir (LDV)

Ledipasvir, the first NS5A inhibitor approved by the FDA in 2014 for treating HCV genotype 1, is available only as an FDC with sofosbuvir [99]. The treatment is administered for 12 weeks in patients without previous treatment, without cirrhosis or compensated cirrhosis, reporting SVRs greater than 90% [99]. In HIV-coinfected patients, treatment schemes for 12 weeks with sofosbuvir/ledipasvir provided an SVR of 96%. Similarly, this therapeutic regimen was approved as the initial treatment in paediatric patients, as sofosbuvir/ledipasvir is well tolerated and highly effective in children between 3 and 6 years of age with chronic HCV infection. Phase 3 studies showed that less than 1% of subjects discontinued treatment due to side effects [100].

### 3.3.2. Daclatasvir (DCV)

Another NS5A inhibitor approved in 2014 by the European Union and, later, in 2015 by the FDA was daclastavir [101]. Its use was indicated in conjunction with sofosbuvir for many years because of its synergistic antiviral activity for HCV genotypes 1 and 3 [102]. Cytochrome P450 3A4 metabolises the compound, and a dose adjustment was required when used with drugs that significantly affect these enzymes, such as certain antiretrovirals [103].

It was used to treat chronic HCV genotype 1 and 3 infection, asuprenir, and PEG-IFN-RBV [75]. The RSV rate was 94% for genotype 2 and 90% for genotype 3 when combined with sofosbuvir or velpatasvir/sofosbuvir. Daclatasvir is not currently included in HCV treatment regimens [75].

### 3.3.3. Ombitasvir

By 2014, ombitasvir, an NS5A protein inhibitor, was approved to treat HCV genotype 1, and in 2015 it was approved for genotype 4 as part of the paritaprevir/ritonavir/ombitasvir FDC [104]. It was used as therapy for HCV-infected patients who did not respond to the first protease inhibitors [104]. Several studies have demonstrated its high efficacy in patients infected with genotype 1 who received previous treatment and those infected with HIV. Ombitasvir is discontinued and has been replaced by pangenotypic antivirals [75].

### 3.3.4. Elbasvir

Approved by the FDA for 2016 was elbasvir, an NS5A replication complex inhibitor with broad genotypic activity for use in combination with grazoprevir in infections with HCV genotypes 1 and 4, as well as in patients who have failed to achieve SVR with previous DAA regimens [105]. The combination of grazoprevir and elbasvir was active against NS3/4A protease inhibitor resistance in vitro and in vivo for genotype 1b [106]. No dosing modifications are required for renal or mild to moderate hepatic impairment patients. Side effects are similar to those reported for grazoprevir [75].

### 3.3.5. Velpatasvir and Pibrentasvir

Velpatasvir was approved by the FDA in 2016 for use alongside sofosbuvir for treating patients infected with genotypes 1 to 6 [107]. High SVRs upon veltaspavir treatment were observed for all the genotypes, except in patients infected with genotype 3, where RBV was incorporated into the treatment. Currently, the ASDL/IDSA guidelines recommend 12 weeks of treatment with sofosbuvir (400 mg)/velpatasvir (100 mg) for previously untreated adult patients and paediatric patients older than 3 years of age infected with

HCV [108]. Similarly, velpatasvir is used in conjunction with sofosbuvir and voxileprevir in cases of treatment failure, specifically with sofosbuvir and glecaprevir/pibrentasvir regimens [109]. The combination of glecaprevir/pibrentasvir has been highly effective in drug users, with an SVR of 93%. In other patients, the SVR was around 95% [110].

Figure 2 presents a historical overview of the treatments used in HCV.

## 4. Treatment in Coinfection (HCV/HIV, HBV/HCV, HBV/HCV/HIV)

There are approximately 2 million people worldwide who have HIV/HCV coinfections; most of them (1.3 million) are parenteral drug users. The risk of HCV infection in HIV patients is six times higher than in the general population. It is recommended to routinely test for HCV infection in persons with recent HIV infection due to the similarities in infection routes and the increased risk of liver pathology compared to patients without HIV infection [48,49].

The current IDSA guideline recommendations suggest that patients living with treatment-naïve HIV and HCV coinfection, regardless of liver involvement (with or without cirrhosis), are good candidates for DAAs based on the results of the phase 4 MINMON (minimal monitoring approach) clinical trial conducted by Solomon et al. in 2022 [111]. The treatment scheme involved sofosbuvir and velpatasvir, with an SVR of 95%. DAA treatment benefits patients with HIV infection while promoting HCV clearance [112].

In HBV/HCV coinfection, defining the genotype of HCV causing the infection is essential since the coinfection can be HCV-dominant, in which DAA drugs will provide a clear rationale [113]. It is vital to monitor circulating HBV DNA even with the nonexistence of antibodies and circulating HBV core protein [113]. Immunocompromised individuals should be closely monitored to avoid viral escape and hepatic disease [113]. PEGIFN-RBV can be added to prevent HBV replication to decrease chronic infection and liver damage. There is a risk of HBV reactivation after DAA treatment in coinfected patients [114,115]. The HBV core antigen and DNA should be closely monitored, and PEGIFN-RBV therapy should be used if required to avoid cirrhosis and the occurrence of hepatocarcinoma [115].

There is little information in the literature about coinfection with the three viruses, probably because of the lack of detectable antibodies against HBV proteins in coinfections [48]. HIV patients treated with triple therapy for an extended period may be partially protected from new HCV or HBV infections [48,49]. In naïve patients, active HCV and HBV replication may induce early liver damage. In HBV/HIV coinfection, tenofovir and entecavir are recommended to block HBV and HIV replication [48,49]. These drugs may benefit patients with triple infection after DAA treatment has concluded. Recently, Cairoli et al. [116] and Mirzaei et al. [117] have reported the importance of miRNA in coinfected individuals. Future research should focus on potential biomarkers to facilitate infection screening and response to therapy.

## 5. Treatment of Pregnant Women, Vertical Transmission and Paediatric Care

Vertical transmission of HCV occurs in 5.8% of infants from HCV-infected women and up to 12% of HIV/HCV coinfected women [118]. Reports published by the CDC in 2016 revealed the increasing risk of perinatal HCV transmission in specific high-risk areas of the United States. The HCV incidence has increased among young adults and women of childbearing age in these areas [118–121]. In the absence of an HCV vaccine, there is an immediate need to improve the availability of HCV screening among at-risk individuals, including children born to HCV-infected mothers [119–121].

The USPSTF (United States Preventive Services Task Force) and the CDC issued recommendations in 2020 concerning the importance of HCV screening in pregnant women at the start of prenatal care and during pregnancy and in those individuals undergoing fertility treatment [119]. It is also suggested to request HCV tests in sperm and ovule donors. Antiviral therapy is recommended before pregnancy is considered. Caesarean delivery is not recommended to prevent perinatal transmission [119]. Breastfeeding is not

contraindicated, except in the context of an HIV-coinfected mother [119–121]. No large-scale studies have been conducted to evaluate the safety of DAAs during pregnancy, and some groups suggest using DAAs during pregnancy on a case-by-case basis [122,123]. However, the Maternal-Fetal Society of the ACOG (American College of Obstetrics and Gynecology) recommends that AAD regimens should only be used in the context of a clinical trial or that antiviral treatment should be deferred until the postpartum period, as they are not currently approved for use in pregnancy [124]. They also suggest measures to reduce the risk of transmission during delivery by recommending avoiding internal foetal monitoring, prolonged rupture of membranes and episiotomy [124]. An open-label phase 1 clinical trial evaluated the use of ledipasvir/sofosbuvir in pregnant women between the second and third trimesters of gestation through pharmacokinetic studies, concluding that the treatment was safe and effective [125]. The study included only 29 pregnant women, a limited population to extrapolate the results on a large scale [125]. There is a need for increased research on antiviral therapies in pregnant women.

HCV infection in children and adolescents is a critical problem in underdeveloped countries. It has been estimated worldwide that 3.5 million children and adolescents are chronically infected with HCV [126,127]. Treatment of the paediatric population based on adult pharmacokinetics has been rationalised by adjusting the adult successful treatment schemes [126–129]. It is important to note that parental HBV and HCV infection may be risk factors for hepatic and non-hepatic cancers in children [130]; therefore, ensuring that infected children are treated as soon as possible is critical. In HCV–HIV coinfection, there is a high risk of vertical transmission and a high possibility of chronic liver disease due to an immature immune response. Research in paediatric coinfection is urgently needed.

## 6. Host Genetics, Infection and Response to HCV Treatments

Several studies have found an association between the host genetic factor, spontaneous clearance of the virus, the treatment response and the risk of fibrosis and/or hepatocarcinoma. Most studied single nucleotide polymorphisms (SNPs) are from the IL28B gene; however, several other SNPs in different genes have been involved: interferon-λ3, interferon-λ4, IL-12, TNF, IL-10 and Toll-like receptors 3 and 9 [131–146]. Moreover, a relationship has been recently shown between increased mortality and epigenetic changes related to age in a group of intravenous drug users coinfected with HCV–HIV [147]. These results suggest that epigenetic modifications may also be relevant to the infection and may jeopardise the effectiveness of the treatment.

Several SNPs near the IL28B gene have been related to the spontaneous resolution of HCV infection [131]. In addition, the SNPs rs12979860 and rs8099917 are strongly associated with the response to treatment with PEGIFN/RBV [132–134]. Patients with the homozygous CC genotype achieve a higher SVR with PEGIFN/RBV treatment than patients with the TT or CT genotypes [134]. A significantly higher SVR was reported for the TT homozygous allele of the IL28B gene rs8099917 compared to the other genotypes [135].

Four SNPs, rs12979860 and rs8099917 for IL28B and rs368234815 and rs117648444 for IFN-λ4, have been associated with HCV susceptibility to infection and response to treatment [135–137]. Ge et al. [132] showed that in the rs12979860 SNP, the favourable allele (CC genotype) was reported in Caucasians who responded positively to IFN treatment. In contrast, the unfavourable TT genotype was more common in African Americans [132]. African Americans with the CC genotype of the SNP rs12979860 responded better to treatment, with a higher SVR, than Caucasians with the TT genotype [132].

Patients with the SNP rs12979860 CC polymorphism are more likely to achieve an SVR at 12 weeks of treatment with sofosbuvir/daclastavir in genotype 4-infected Egyptian patients [136]. This gene has been studied both as a predictor of response and as a factor of resistance to treatment in patients treated with sofosbuvir, raising the possibility of performing genetic analysis for the SNP in IL-28B before initiating antiviral treatment [139,140].

There is a significant association between HCV infection and coinfection with genetic polymorphisms of Toll-like receptors [148–150]. The allele of the TLR3 rs13126816 decreased

the odds of a virologic response to HCV therapy in HCV/HIV coinfected patients [149], while the TLR9 rs352140 (G/A variant) may be necessary in HCV and HBV/HCV coinfection [148].

The polymorphisms of interferon regulatory factors have not been studied in detail. Talaat and coworkers [151], in a small group of Egyptian patients, have shown that the IRF3 SNP polymorphism rs2304204 (−925A/G) is a protective genotype for liver cirrhosis. Based on the antiviral effect of the IFN pathways, it is crucial to analyse the genetic polymorphism, which may be more prevalent in the infected population and a risk factor in endemic areas.

The miRNAs can also be considered key players in the antiviral response. New reports on miRNA have provided insights into single and multiple infections and possible targets for therapy [116,117,152,153]. However, large-scale studies are required to validate the miRNA signatures in HCV-infected and coinfected individuals and their response to treatment.

### 7. Resistance-Associated Substitutions (RAS)

The main challenge with DAA treatment is resistance-associated substitution (RAS), the leading cause of DAA resistance [154–156]. This treatment resistance is an inevitable and intrinsic problem, as HCV is highly adaptable [154–156]. These mutations are due to the lack of correction of HCV polymerase, leading to base mutations for each viral replication [154–156]. SARs occur because of different treatment regimens, the various HCV genotypes and subtypes, and the geographic distribution. SARs in other regions of the NS3/4 protease (F53S, Q80K/R, S122R, R155K, A156T/V, D168) are important causes of treatment failure with protease inhibitors, such as simeprevir, grazoprevir, asunaprevir, and paritaprevir [157].

On the other hand, NS5A inhibitors are an indispensable component of all first-line DAA regimens, making them the class of HCV drugs where resistance is most relevant. RASs found after NS5A inhibitors (M28A/G/T, Q30E/H/R, L31F/M/V, P32L/S, H58D, Y93H) are more frequent in HCV genotype 1b and genotype 3. The presence of these RASs in NS5A was demonstrated to show a decrease in the SVR in the first week of antiviral treatment with sofosbuvir/daclastavir, especially in HCV genotype 3 [157,158].

The spectrum of mutations associated with NS5B inhibitors is similarly broad. Because all nucleoside inhibitors target the highly conserved active sites of polymerases, these inhibitors tend to be pan-generic. However, RASs such as S282T have been identified to decrease the efficacy of sofosbuvir and the C316N mutation that reduces dasabuvir's effectiveness [155,157,158].

Even though the number of patients with RASs has not been reported as highly significant, and the consensus is a change of therapy if an SVR is not achieved, there is a need for good epidemiological studies involving patients infected only with HCV and coinfected to determine the actual extent of the RASs. These studies should not limit the generation of more specific and effective compounds for treatment.

### 8. Other Strategies

The World Health Organization (WHO) aims to achieve a ~90% reduction in new HCV infections by 2030 [159]. Treatment efficacy has improved since the introduction of DAAs, achieving up to a 95% pangenotypic cure; however, there are still challenges, especially in high endemic areas. There is a need to develop preventive and therapeutic HCV vaccines; the candidate vaccines studied have shown limited efficacy due to low immunogenicity [160–163]. There is a debate about whether B-lymphocyte and T-lymphocyte responses are necessary to develop an effective vaccine [164]; however, new strategies to develop a specific and long-lasting vaccine have improved after the experiences encountered with the SARS-CoV-2 virus [165,166]. A protein-based HCV vaccine could induce cell-mediated and humoral immunity; non-structural proteins and HCV E1/E2 proteins [167,168] seem promising. Other strategies based on the spontaneous clearance of the virus may be used. Recent studies have shown that combining several adjuvants can be useful in increasing the efficacy of vaccines via the induction of several receptors of innate immunity. An example of this is the study of HCV-related immunoadjuvants,

where an emulsion (MF59), lipid-based nanoparticles (archaeosomes) and a combination delivery immunostimulating system, Alhydrogel-MPL, using recombinant HCV E1E2 glycoproteins, were compared. All the formulations with adjuvants showed enhanced immunogenicity with significant neutralisation activity compared to antigen alone; however, no cellular response was detected for the formulation with the MF59 adjuvant [168].

On the other hand, a study by Lin et al. [169] evaluated the use of recombinant HCV polypeptides combined with various Th1-type adjuvants and replication-defective alphaviral particles encoding HCV proteins in various priming/boosting modalities in BALB/c mice. Chimeric alphaviral-defective particles derived from Venezuelan equine encephalitis virus and Sindbis encoding the gpE1/gpE2 heterodimer of the HCV envelope glycoprotein (E1E2) or non-structural proteins 3, 4, and 5 (NS345) elicited a robust $CD^{8+}$ T cell response but a low $CD^{4+}$ helper T cell response to these HCV gene products. In contrast, recombinant E1E2 glycoproteins with MF59 adjuvant containing a CpG oligonucleotide elicited strong $CD^{4+}$ helper T cell responses but no $CD^{8+}$ T cell responses. A recombinant NS345 polyprotein also stimulated strong $CD^{4+}$ T lymphocyte responses but no $CD^{8+}$ T cell responses when used with ISCOMATRIX™ containing CpG. Optimal $CD^{4+}$ and $CD^{8+}$ T cell responses against E1E2 and NS345 were achieved by sensitising with Th1 adjuvants and then boosting with defective chimeric alphaviruses expressing these HCV genes. Therefore, these authors concluded that the formulation of this vaccine and the regimen used may be effective for treating HCV. Therefore, these authors concluded that the formulation of this vaccine and the regimen used might be effective in humans for protection against this highly heterogeneous virus [169].

The failure of a T cell vaccine based on the use of viral vectors expressing HCV non-structural protein sequences to prevent chronic hepatitis C has indicated that the induction of neutralising antibodies (NAb) should be essential in future vaccines. Finally, the Nab generated by vaccines should contain the main target, which are the structural proteins, including the HCV envelope glycoproteins (E1 and E2) [170].

Another strategy of interest in HCV therapeutics has been the use of checkpoint inhibitors previously used in cancer treatments. This approach has begun to be used as a treatment in patients with hepatocarcinoma and untreated HCV infection, showing no toxicity in using checkpoint inhibitors, thus opening up a range of possibilities for untreated and cured HCV oncology patients [171].

Treatment of acute HCV infection in patients undergoing organ transplantation has not been adequately explored [172]. It is clear that also there is a risk of infection from the donor, which may jeopardise the recipient's response. In addition, there is a lack of data regarding the use of DAAs in managing chronic HCV-infected patients with HCV superinfection. In HCV superinfection, there is a potential genotype switch and mixed viral strain infection in patients with HCV superinfection; pangenotypic DAA regimens are the preferred treatment choices to secure satisfactory viral eradication. It is essential to clarify that the risk of reinfection after successful DAA treatment is low; however, due to the possibility of the virus infecting non-hepatic cells, the chance of tissue niches should not be disregarded.

In patients undergoing solid transplants, monoclonal therapy blocking the possible entry of the virus into the cells has been explored [172–174]. Since the monoclonal antibody against CD81 can be used to treat colorectal, liver and gastric cancers [174], it can be used in transplanted patients.

HCV can infect hepatic and non-hepatic cells via the lipid receptors, the scavenger receptor B type I, the LDL receptor, the ApoE receptor and glycosaminoglycan or heparan sulphate proteoglycan [175]. Lipid-lowering drugs, including ezetimibe, have been suggested to inhibit the infection of the virus in the cell and, in conjunction with other therapies, have been tested in chronic patients and patients at risk [175]. It is assumed that no well-designed clinical trial has been posted, that the inhibition of cholesterol synthesis will decrease the possibility of internalisation of the complex, and that it will also affect the lipid moiety of the cell capsid [175]. Ezetimibe seems to be more effective, along with DAAs,

in achieving this objective [175,176]. Several clinical trials have been posted concerning the use of (1) fluvastatin and simvastatin to improve IFN or PEGIFN sensitivity (NCT01377909); (2) statins to potentiate the effect of the DAA combination sofosbuvir/daclatasvir (Egypt) NCT03490097; (3) the trial NCT00487318, a randomised control study, including genotypes 1 and 3, in which statin is added to the combination sofosbuvir/daclatasvir/ribavirin to test if statin potentiates the antiviral effect; (4) the trial NCT00926614 (San Antonio, TX, USA) aimed to study the effect of an insulin-sensitising thiazolidinedione plus a atorvastatin in improving sustained virologic response rates in patients who have previously not responded or relapsed on standard PEGIFN and ribavirin therapy; and (5) the use of different doses of rosuvastatin and atorvastatin to determine if these compounds affect the HCV viral load and liver parameters (NCT00446940). However, there is no single and reliable publication on these trials.

Recently, it has been shown that heparanse-1 is upregulated by HCV infection and favours its replication [177]. Heparanase-1 has been involved in tumour growth [178], and it may be related to hepatocarcinoma-induced HCV infection. Further studies are required in this area.

A recent trial has shown that erlotinib treatment is safe in noncirrhotic CHC patients. Antiviral activity at 100 mg/d confirms the functional role of EGFR as an HCV host factor in patients [179]. The inhibition of viral entry with dasatinib was also described, and the role of EphA2 was then confirmed [175]. In a similar fashion, the inhibition of claudin-1 by monoclonal antibodies has been studied [175], and just recently, a new humanised version has been described [180] but no results of trials have been published.

RNA interference (RNAi) gene silencing and antisense oligonucleotide suppressive functions have been described. The most promising therapy involves miRNA-122 induction to inhibit viral entry and blocking miR-155 to prevent hepatocellular carcinoma after HCV infection [181].

## 9. Conclusions

According to the WHO, there are about 1.5 million new HCV infections annually and an estimated 8 million chronically infected people worldwide. Even though most individuals clear the virus, many individuals require appropriate therapy. Therapies have evolved from IFN, pegylated-IFN, and RBV to the DAAs, significantly decreasing the viral burden in infected patients. However, there are significant limitations. Simple and economical diagnostic tests are required in health centres with high incidences of infections, generally in underdeveloped countries. Early detection reduces the epidemiological transmission of the virus and decreases the risk of developing chronic hepatitis.

Mohamed and coworkers showed an increased risk of colorectal, pancreatic, lung, and breast cancer development in 1476 HCV patients compared to 1550 controls. In addition, in the HCV-infected group, the colorectal and pancreatic cancer survival was significantly less than in the controls, but not in the lung and breast cancers [182]. Questions now arise concerning the relationship between extrahepatic cancer and chronic infection.

The emergence of RASs puts pressure on developing new antiviral treatments with high efficacy, an SVR, and low rates of adverse events. There is a need for an effective vaccine to minimise infection and protect vulnerable populations, especially in underdeveloped countries. There is also a need for an epidemiological follow-up of treated populations to analyse the effectiveness of treatments and medical attention. Any effort to cure and protect individuals from this infectious disease is worthwhile. Figure 3 summarises all of the therapies discussed in the text.

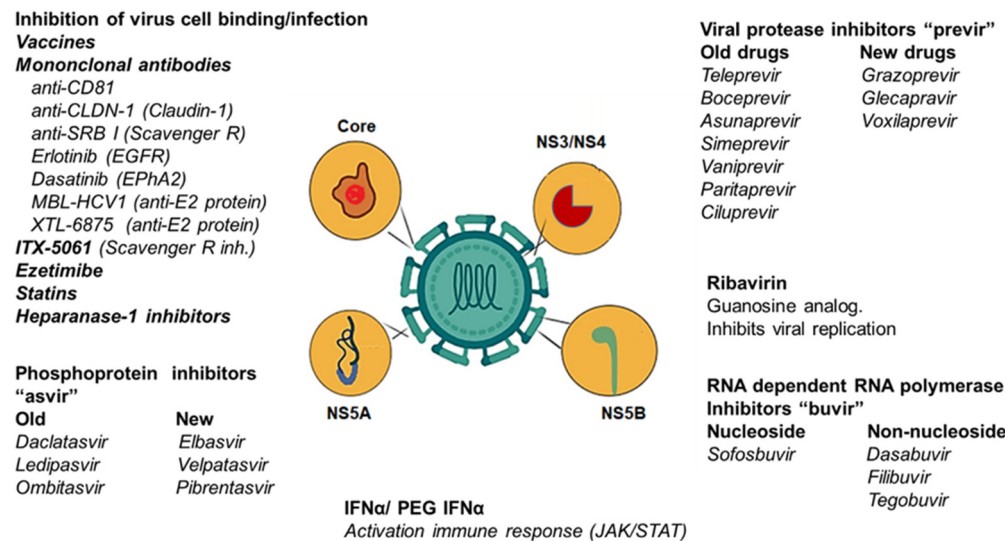

**Figure 3.** New and old therapeutic strategies against HCV infection. The figure represents therapeutic targets, from cell binding/infection to general antiviral drugs such as IFNα. The old and new inhibitors of NS3/NS4, NS5A and NS5B are represented. Several vaccine strategies are being tested. Statins, ezetimibe and other lipid-lowering drugs have been tested in clinical trials; however, no results have been posted. ITX-5061 (an antagonist of scavenger receptor B1) was evaluated for safety (clinical trial phase I NCT01165359 in HCV naïve patients and NCT01560468 in liver transplant patients). Ezetimibe has been used in transplant patients from HCV-positive donors (clinical trial NCT04017338). Other inhibitors have either been tested in clinical trials or new schemes have been developed to undergo clinical trials.

Several areas require research. The difficulties associated with eradicating HCV and HBV infection in endemic areas are complicated; new genotypes may be developed, complicating the treatment and medical care, the management of transplanted patients and chronic patients, especially elders, with low therapeutic options and, finally, the development of protective vaccines that will be a breakthrough, especially for underdeveloped countries.

**Author Contributions:** Conceptualisation, C.M., A.H.G., F.I.C., F.I.T. and S.J.M.; investigation, C.M., F.I.C. and J.B.D.S.; writing—original draft preparation, C.M., F.I.C. and S.J.M.; writing—review and editing, A.H.G. and J.B.D.S.; visualisation, A.H.G., F.I.C. and J.B.D.S.; supervision, J.B.D.S.; project administration, A.H.G.; funding acquisition, A.H.G. All authors have read and agreed to the published version of the manuscript.

**Funding:** This work was financed by the National Fund for Science, Technology, and Innovation of Venezuela (FONACIT), an entity attached to the Ministry of Popular Power for Science and Technology of the Bolivarian Republic of Venezuela (MINCYT). C.M. and F.I.C. are fellows of the programme. A.H.G. is the responsible person. J.B.D.S. is partially financed by the National Institute of Virology and Bacteriology (Programme EXCELES, ID Project No. LX22NPO5103)—Funded by the European Union—Next Generation EU from the Ministry of Education, Youth and Sports of the Czech Republic (MEYS).

**Institutional Review Board Statement:** Not applicable.

**Informed Consent Statement:** Not applicable.

**Acknowledgments:** The authors would like to thank Mercedes Zabaleta, Director of the Institute of Immunology, Universidad Central de Venezuela, and Leopoldo Deibis for their support.

**Conflicts of Interest:** The authors declare no conflict of interest.

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
