# Peer review of "A Synopsis of Hepatitis C Virus Treatments and Future Perspectives"

_cimb, doi:10.3390/cimb45100521_

Round 1

Reviewer 1 Report

The review A SYNOPSIS OF HEPATITIS C VIRUS TREATMENTS AND 2 FUTURE PERSPECTIVES contains all the necessary information about the current available therapy for HCV and some insights to what efforts should be made in the future to continue the successful treatment of the infection. I do, however, have some suggestions for the authors. The introduction part of the manuscript should be divided into more chapters with subtitles to make it more clear for the readers, and perhaps start with viral genome, for one chapter and then continue with the cell entry and host response. Resistance to DAA should also be a separate chapter, while authors mention resistance when describing certain DAA. Other than that I find the manuscript suitable for publishing.

The quality of the English language is fine, only a minor editing is required.

Author Response

We thank the reviewer for the fruitful comments. The introduction was divided as suggested, it is highlighted in yellow. The part of DAA resistance was separated in section 7 in the original manuscript, and we kept the text accordingly.

Reviewer 2 Report

The paper presented to me for review is a very skillfully and transparently written review on the current and future treatment HCV. The authors analyzed the current and the newest literature. The paper undoubtedly provides an important summary and aggregate presentation of the research findings. It is written in good and understandable language.

I have no major objections to the paper; however, before accepting it for publication, I would suggest a few additions that could improve the quality of the paper:

1. in the introduction it would be appropriate to refer to extrahepatic manifestations of HCV including involvement of the nervous system and in the analysis results to include studies suggesting improvement of bioelectrical and metabolic brain function after DDA treatment in patients based on: PMID: 30769221 and PMID: 30641198 

2. for consideration I leave the presentation of the most important conclusions in the form of at least one summary table to dilute the text a bit and improve the visual aspect of the paper

Author Response

We thank the reviewer for the suggestions.

Concerning point 1. The two references were added in the introduction, and we found an interesting study DOI: 10.5114/ceh.2023.130783 concerning a relationship with cancer incidence that we considered appropriate for the manuscript.

Point 2. 

We thank the reviewer for the comment. However, Figure 3 summarises all the therapies and new approaches; we feel that a table will be repetitive.